# Sustainability Development in Hospitality: The Effect of Perceived Value on Customers' Green Restaurant Behavioral Intention

**Yi-Man Teng [1] and Kun-Shan Wu [2,*]**

[1] College of Commerce, Yango University, Fuzhou 350015, China; yimanteng@gmail.com
[2] Department of Business Administration, Tamkang University, Taipei 25137, Taiwan
* Correspondence: kunshan@mail.tku.edu.tw; Tel.: +886-2-2621-5656#3314

**Abstract:** Recently, sustainability management has developed in the hospitality industry. Green restaurants have been progressively joining the hospitality industry. Consumer patronage determines the sustainable development of the green restaurant. This paper aims to explore the structural relationships among perceived values (hedonic and utilitarian values), consumer preference and behavior intentions of the green restaurant. A total of 278 valid questionnaires were collected, and the partial least squares (PLS) method was utilized to measure and test the research hypotheses. The study presents empirical evidence showing that hedonic and utilitarian values have significantly and positively affected consumer preferences for green restaurants and that utilitarian value and customer preferences have significantly and positively influenced behavior intentions of green restaurant. Furthermore, the results also demonstrate that consumer preferences partially mediate the relationships between utilitarian value and behavior intentions of the green restaurant. Finally, theoretical and practical implications are discussed and suggestions for future research are provided.

**Keywords:** green restaurant; hedonic value; utilitarian value; preference; behavior intention; partial least squares (PLS)

## 1. Introduction

With ecological damage and the worsening of global warming, environmental sustainability awareness has expanded, and modes of green consumption are becoming increasingly popular. As consumer preference for environment-friendly products has increased in recent years, consumer interest in sustainability management when dining out is increasing.

Kasim [1] asserted the hospitality industry affects the environment and it has been acknowledged as a significant emitter of greenhouse gases, especially carbon dioxide. Taking action to reduce pollution of the environment is necessary. Restaurants have a cumulative impact on the economy, society and environment. On the other hand, U.S. restaurant food and drink sales reached 766 billion U.S. dollars in 2016. An increasing number of people like to dine out at restaurants [2], with dine-out expenditures accounting for more than half of household food expenditures [3]. The restaurant sector is not only a major industry but also a major source of greenhouse gas emissions [4]. Badlwin, Wilberforce and Kapur [5] reported that the foodservice industry accounts for approximately 30% of global greenhouse gases. Scholars said that although the hospitality industry is not considered as great a polluter as the chemical or metallurgical industries, the size and rapid growth of the industry indicates environmentally sustainable operations are required [6].

Recently, some researchers have further investigated the complex relationship between consumer behavior and sustainability management practices in the hospitality industry, e.g., sustainability



management practice regarding organic and local food, green design, reduced energy and water consumption, and the sourcing of sustainable fitting and fixtures [6]. The sustainability management practice characteristics are similar to the green restaurant. Jang et al. [7] define the green restaurant as "a restaurant offering a selection of sustainable food menu items that include locally grown or organic food, that adopts a recycling program, that uses energy efficiently and that makes a concerted effort to limit solid waste". Compared to standard restaurants, sustainable restaurants such as green restaurants emphasize the three Rs (recycle, reuse and reduce), the two Es (efficiency and energy) and the development of green food menus [8,9].

Some scholars pointed out green restaurants have been increasing [10]. Increasing numbers of green restaurants are entering the market [11]. Jang et al. [7] claimed that even though 63 percent of respondents said they have never visited a green restaurant, they are willing to spend more money on a green restaurant. Hence, green dining has become an important subject for the hospitality field scholars, restaurant owners and managers [4,9,10,12,13].

Research on green restaurants has been conducted from various perspectives, such as based on consumer perceptions and beliefs about green consumption, and based on the individual characteristics of green restaurants [4,14–16]. Some researchers have examined the sustainable attributes of green restaurants [17]. A growing number of hospitality scholars has focused on levels of quality among green restaurants [11,18]. However, to our knowledge, the effect of perceived value on green restaurant patronage intentions has rarely been explored. Perceived value plays a key role in describing the psychological state of the customer while selecting a product or service. Several articles have focused on how hedonic and utilitarian value affect consumption behavior [19–21]. In hospitality research, some studies have also adopted the hedonic and utilitarian value and based on the structural equation model (SEM) to test effects on restaurant patron behavioral intentions in relation to different types of restaurants, such as fast-food [21–24], fast-casual [22,25], and luxury restaurants [26]. There is currently limited literature on the dining behaviors of green restaurant consumers.

In the green restaurant environment, hedonic value is a reflection of the customer's emotions in which the customer tends to maximize their feelings of fun, joy, etc. Conversely, utilitarian value refers to the green restaurant customer who view products or services as more practical and necessary. In summary, when a customer uses a service or product, it can be said that hedonistic and utilitarian value refers to the rational or emotional embodiment of the value of a service or product [27]. Hence, there is a need to determine the impact of hedonic and utilitarian value on behavior intentions in the setting of the green restaurant.

Accordingly, to address gaps in the research of previous studies, the present article aims to explore how utilitarian and hedonic values affect Taiwanese consumer preferences and their behavior intentions within the settings of the green restaurant. The partial least squares (PLS) method was utilized to measure and test the research hypotheses in this study.

## 2. Literature Review and Hypotheses Development

### 2.1. Hedonic and Utilitarian Value

Consumer choice is the product of multiple values perception. Perceived value is an important factor in understanding consumer behavior [28]. Among many value theories, the concepts of hedonic and utilitarian value have been widely explored in hospitality studies [20,22,23,26,29–31]. Ryu et al. [24] claimed that utilitarian and hedonic values represent the basis of consumers' evaluations of their experiences, as these two levels of value can explain the most basic potential consumption phenomena. Therefore, through these two levels, the value of consumers can be more fully presented. The nature of green restaurants also suggests that they are likely to offer both hedonic (experiential) and utilitarian (functional) values.

Hedonic value refers to the value-conscious experience obtained from the emotional, social, reputational and aesthetic aspects of a product [32]. Thus, hedonic value is associated with consumers'

desires for entertainment, enjoyment, fun, novelty and excitement [26,30,33]. On the other hand, utilitarian value represents an overall assessment of product value, including its economic, quality value and functional characteristics [32]. As rational problem solvers, utilitarian consumers concentrate on the practical advantages and empirical convenience afforded by an item [34].

The green restaurant provides hedonic value by evoking responses through social or interpersonal experiences and utilitarian value by satisfying physical needs (for nourishment, health, and sustainability). Thus, the decision to patronize a green restaurant is likely to be based on a consumer's evaluation of tangible and intangible considerations, as well as emotional costs and benefits. Conversely, perceived utilitarian values of green restaurant patronage may depend on whether the need spurring patronage has been met. The utilitarian value derived from green restaurant patronage can include consuming fresh organic food for health benefits or satisfying desires for sustainability on the premise that such consumption limits environmental degradation. Green restaurants not only provide sustainable organic food from local farms but also in the form of environmental protection and energy saving operation. These restaurants satisfy consumers' utilitarian values regarding health and sustainability.

### 2.2. Effects of Hedonic and Utilitarian Value on Preferences

Consumer values have shown to positively influence consumer preferences and satisfaction [35,36]. Some researchers believe that consumer values are linked to consumer preferences as well as future consumer intention [37,38]. In some consumer studies, the role of preference seems to play an important part in future behavior intention [39]. Many previous studies have applied the hedonic/utilitarian scale to measure consumer attitudes toward hospitality and tourism products in terms of reliability and validity. For example, one study used the efficacy of the scale to measure festival attendance patterns [20]. Those results were in parallel to other scholars, including studies that verified that the utilitarian and hedonic values were important in consumer preference and future intention of online shopping and in discount sector consumer behavior [19,40]. Recently, Nili et al. [41] indicated that hedonistic and utilitarian values positively influence the preferences and future shopping intention. From the preceding discussion, the following hypotheses are proposed.

**H1:** *The hedonic value of the dining experience has a positive influence on consumers' preferences for green restaurants.*

**H2:** *The utilitarian value of the dining experience has a positive influence on consumers' preferences for green restaurants.*

### 2.3. Effects of Utilitarian and Hedonic Value on Behavior Intentions

In the field of restaurant marketing, considerable research work has suggested and confirmed the causal relationship between perceived value and behavior intentions. For example, Park [33] underlined the significance of the hedonic value of a consumer's experiences with Korean fast food restaurants. Similarly, Ha and Jang [30] evidenced that when patronizing Korean restaurants, the influence of utilitarian value on American customers' levels of satisfaction and behavior intentions is stronger than the corresponding hedonic value. Ryu et al. [24] conducted a study of a fast food restaurant in the United States showing similar results. Similar results have been identified from studies of fast-food [21] and fast-casual restaurants in Iran [25].

In reference to the chain restaurant environment, Hyun et al. [42] revealed that perceived values significantly impact behavior intentions. These studies also demonstrated that the utilitarian value has a greater effect on behavior intentions than the hedonic value. While investigating the relationship between fast food chain restaurant attributes, hedonistic and utilitarian values, and behavioral intentions among Taiwan's Y-generation consumers, Chiang and Li [23] found that hedonic values affected behavioral intentions more than utilitarian value did. Basaran and Buyukyilmaz [22] compared

the impacts of hedonistic and utilitarian values on young consumers' levels of satisfaction and behavior intentions in reference to the fast-food and fast-casual restaurant sectors. They discovered that both hedonistic and utilitarian values played important role on behavioral intentions related to fast-food and fast-casual restaurants. In addition, the utilitarian value has a greater effect on behavior intentions as compared to the hedonic value for the fast-food restaurant sector while the impact of the hedonic value on behavior intentions is greater than that of the utilitarian value for the fast-casual restaurant sector. In reference to luxury restaurants, Hyun and Park [26] examined the antecedents and consequences of travelers' unique needs from their restaurant experiences. Their results showed that perceived value (hedonic and utilitarian value) significantly influences their behavior intentions. These researchers also emphasized that utilitarian values have a greater effect on behavioral intentions than hedonistic values. Based on the above-mentioned literature review, we proposed the following assumptions.

**H3:** *The hedonic value of the dining experience has a positive influence on consumers' behavior intentions to visit green restaurants.*

**H4:** *The utilitarian value of the dining experience has a positive influence on consumers' behavior intentions to visit green restaurants.*

### 2.4. Effects of Preferences on Behavior Intentions

Many studies have shown that preferences have a significant influence on behavior intentions, on the willingness to purchase, and on word of mouth processes in a variety of shopping environments [19,36,39,43–45]. Preferences also affect levels of satisfaction, loyalty, and purchasing/repurchasing behaviors of retail patronage intentions [39,45,46]. Despite the fact that Fishbein and Stasson [47] declared that intentions are by nature motivational, intentions may not be triggered when preferences are not presented. Bagozzi [43] also argued that preferences are different from intentions and even declared that intentions may not be activated unless preferences are present. Building on this, we propose that consumers' preference for green restaurants will positively impact consumers' willingness to patronage green restaurants.

In addition, Nili et al. [41] examined effects of hedonistic and utilitarian values on the preferences and intentions of Iran consumers in the online shopping setting. Their results implied a significant relationship between judgments of hedonistic and utilitarian values and preferences. Furthermore, preferences have a significant relationship with future purchasing intentions. Rahman and Abdel Fattach [48] explored the associations among service quality, tourist preferences, satisfaction and intentions to patronize a restaurant for food items in Malaysia. Their study also showed that tourists' preferences have a significantly positive impact on tourists' intentions in selecting a specific restaurant. Based on the preceding discussion, the assumption as follows.

**H5:** *Preferences for green restaurants have a positive influence on the behavior intentions of consumers.*

Based on the preceding discussion, the model presented in this paper is shown in Figure 1.

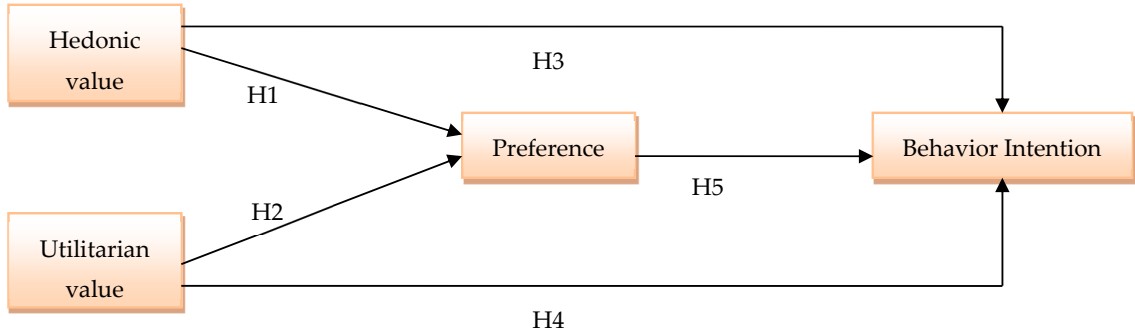

**Figure 1.** Conceptual model.

## 3. Methodology

### 3.1. Data Collection Process

The data collection process follows. An empirical study was conducted using convenience sampling of customers who dined at a large shopping mall in Taipei City, Taiwan. At the beginning of the survey, participants were asked to indicate their age, and only customers who were 20 or older were included in this study. As a result, a total of 300 questionnaires were collected. After eliminating questionnaires with errors and/or missing values, 278 valid responses were utilized for data analysis. The response rate of 92.67% indicated that the sample selection bias was not a concern, as higher the response rate lowered the sample selection bias [49,50].

### 3.2. Instrument Development and Measurement

The original questionnaire was written in English, then translated according to the Douglas and Craig [51] recommendation for the native speaking (Chinese), in order to ascertain the cultural equivalence.

The study instrument was developed by adapting the items from prior literatures that used well-established scales. All major scale items adapt 7-point Likert scale measurement, ranging from 1 (strongly disagree) to 7 (strongly agree). For this study, we adapted ten items for hedonic and utilitarian values from Voss et al. [52]. An example of a hedonic value question was "For me, patronage of a green restaurant is delightful". An example of a utilitarian value question was "For me, patronage of a green restaurant is functional". Similarly, three items for preference were adapted from Overby and Lee [19] that were relevant to the study's context. An example of a question was "I prefer the green restaurant to other restaurants of its type". The authors measured the seven-item behavior intention scale adopted from [14,19]. An example of a question was "I will choose a green restaurant with my friend when dining out". The measures for each variable are presented in Table 1.

### 3.3. Common Method Bias

Data on the constructs considered in this study were collected from the same respondent, and common method bias problems were difficult to avoid. Therefore, we used Harman's single-factor test to examine whether levels of common method bias in the dataset were pronounced [53]. The first factors explained 33.45% of the total variance, which is less than the benchmark of 50% set by Podsakoff and Organ [54] indicating no obvious common method bias in our dataset.

**Table 1.** Statistics of variables.

| Constructs/Items | Standardized Factor Loading | t-value | CR | AVE | Cronbach's α |
|---|---|---|---|---|---|
| Hedonic value (HV): For me, patronizing a green restaurant is | | | | | |
| HV1: exciting | 0.887 | 61.057 ** | | | |
| HV2: delightful | 0.899 | 66.697 ** | | | |
| HV3: fun | 0.898 | 57.149 *** | 0.943 | 0.767 | 0.924 |
| HV4: thrilling | 0.885 | 57.350 *** | | | |
| HV5: exciting | 0.808 | 22.296 *** | | | |
| Utilitarian value (UV): For me, patronizing a green restaurant is | | | | | |
| UV1: Effective | 0.876 | 47.984 *** | | | |
| UV2: Helpful | 0.871 | 48.460 *** | | | |
| UV3: Functional | 0.869 | 51.958 *** | 0.929 | 0.723 | 0.903 |
| UV4: Necessary | 0.746 | 15.317 *** | | | |
| UV5: Practical | 0.882 | 58.390 *** | | | |
| Preference (PR) | | | | | |
| PR1: In regard to dining out, a green restaurant is my first preference. | 0.892 | 58.586 *** | | | |
| PR2: I prefer a green restaurant to another restaurant of the same type. | 0.909 | 67.556 *** | 0.930 | 0.817 | 0.888 |
| PR3: I consider the green restaurant to be my favored type of restaurant. | 0.910 | 80.138 *** | | | |
| Behavior intentions (BI) | | | | | |
| BI1: I am willing to patronize a green restaurant when dining out. | 0.638 | 3.437 *** | | | |
| BI2: I plan to eat at a green restaurant when dining out. | 0.851 | 41.066 *** | | | |
| BI3: I make an effort to dine at a green restaurant when dining out. | 0.819 | 27.483 *** | | | |
| BI4: I express my intentions to patronize a green restaurant when dining out. | 0.850 | 37.656 *** | 0.945 | 0.714 | 0.925 |
| BI5: I select a green restaurant with my friends when dining out. | 0.890 | 64.251 *** | | | |
| BI6: I intend to continue to dine at green restaurants in the future. | 0.916 | 83.196 *** | | | |
| BI7: In the future, green restaurants will be one of the first choices when I dine out. | 0.919 | 92.533 *** | | | |

Note: CR denotes the composite reliability; AVE denotes the average variance extracted; ** denotes $p < 0.01$; *** denotes $p < 0.001$.

## 4. Results

### 4.1. Descriptive Statistics

In terms of demographics among 278 participants, there were slightly more female respondents in this study (n = 178, 64%) than there were male (36%). In terms of ages, 74 participants were aged between 30–39 (57.3%) and 65 participants were aged between 40–49 (23.4%). In total, 162 participants (58.3%) had an undergraduate degree, and 112 participants (40.3%) indicated that their individual monthly income was between 30,000 and 50,000 New Taiwan dollars.

### 4.2. Measurement Model

The hypotheses proposed in the conceptual model were tested using the partial least squares (PLS) method as such a test places minimum sample size limit of the local measurement scales and residual distributions [55], focusing on each path coefficient, and focusing on the interpretation of variance instead of the overall model fit [56].

From table 1, it can be found that all Cronbach α values were above 0.8 which met the evaluation criteria of Nunnally [57] and Hair et al. [58]. Besides, all composite reliability (CR) values were more than 0.9, which met the criterion of Anderson and Gerbing [59]. The results implied the model having good reliability.

On the one hand, it also can be found that all factor loadings exceed 0.6, and the value of average variance extracted (AVE) for each construct exceeds 0.7 (Table 1), which met the criterion of Anderson and Gerbing [59], Fornell and Larcker [60]. The results display that the model has convergent validity.

Furthermore, the cross-loadings of all items ranked highest among the respective factors (Table 2), and the values of AVE for each construct were greater than the squared correlation between constructs (Table 3), which met the criterion of Fornell and Larcker [60]. The results implied the model having discriminant validity.

**Table 2.** Cross-loadings of constructs.

| Measurement Item | Hedonic Value | Utilitarian Value | Preference | Behavior Intention |
|---|---|---|---|---|
| HV1 | *0.887* | 0.590 | 0.294 | 0.346 |
| HV2 | *0.899* | 0.540 | 0.307 | 0.351 |
| HV3 | *0.898* | 0.615 | 0.289 | 0.311 |
| HV4 | *0.885* | 0.589 | 0.330 | 0.349 |
| HV5 | *0.808* | 0.570 | 0.311 | 0.310 |
| UV1 | 0.640 | *0.876* | 0.337 | 0.388 |
| UV2 | 0.573 | *0.871* | 0.300 | 0.341 |
| UV3 | 0.716 | *0.869* | 0.378 | 0.429 |
| UV4 | 0.303 | *0.746* | 0.249 | 0.272 |
| UV5 | 0.500 | *0.882* | 0.278 | 0.352 |
| PR1 | 0.376 | 0.331 | *0.892* | 0.692 |
| PR2 | 0.320 | 0.358 | *0.909* | 0.707 |
| PR3 | 0.252 | 0.308 | *0.910* | 0.708 |
| BI1 | 0.322 | 0.372 | 0.462 | *0.638* |
| BI2 | 0.272 | 0.335 | 0.687 | *0.851* |
| BI3 | 0.290 | 0.294 | 0.629 | *0.819* |
| BI4 | 0.284 | 0.354 | 0.649 | *0.850* |
| BI5 | 0.323 | 0.368 | 0.683 | *0.890* |
| BI6 | 0.365 | 0.377 | 0.742 | *0.916* |
| BI7 | 0.402 | 0.429 | 0.709 | *0.919* |

**Table 3.** Correlations of constructs and AVE values.

| Construct | HED | UTI | PRE | BI |
|---|---|---|---|---|
| HED | *0.767* | 0.411 | 0.122 | 0.150 |
| UTI | 0.411 | *0.723* | 0.131 | 0.182 |
| PRE | 0.122 | 0.131 | *0.817* | 0.594 |
| BI | 0.150 | 0.182 | 0.594 | *0.714* |

Note: (1) HED: Hedonic; UTI: Utilitarian; PRE: Preference; BI: Behavior intention; (2) Values shown on a diagonal (in bold) denote the AVE values whereas variables shown below the diagonal line denote the squared correlations between each pair of latent constructs.

### 4.3. Structural Model

Empirical results of the structural model and tests of the hypotheses are summarized in Table 4. Of these five hypotheses, four are supported. The bootstrap resampling method with a sample size of 500 was adopted to estimate the theoretical model and hypothesized relationships, generate t values and standard errors, and determine the significance of the path in the structural model.

**Table 4.** Results of the Structure Model.

| Hypothesis | Path Coefficient | Std. Error | t-value | Outcome |
|---|---|---|---|---|
| H1: HED → PRE | 0.189 | 0.065 | 2.898 ** | Supported |
| H2: UTI → PRE | 0.243 | 0.071 | 3.439 *** | Supported |
| H3: HED → BI | 0.042 | 0.038 | 1.118 | Not Supported |
| H4: UTI → BI | 0.136 | 0.055 | 2.489 * | Supported |
| H5: PRE → BI | 0.713 | 0.035 | 20.293 *** | Supported |

Note: (1) HED: Hedonic; UTI: Utilitarian; PRE: Preference; BI: Behavior intention; (2) * $p < 0.05$; ** $p < 0.01$; *** $p < 0.001$.

As is evident from Table 4, the statistical results show that both the hedonic value (β = 0.189, t-value = 2.898, $p < 0.01$) and utilitarian value (β = 0.243, t-value = 3.439, $p < 0.001$) significantly affect consumers' preferences for green restaurants and explaining 15.6% of the variance in preferences. As a result, Hypotheses 1 and 2 are supported. Moreover, utilitarian value is the most important factor influencing preference. In addition, the statistical results indicate that hedonic value (β = 0.042, t-value = 1.118, $p > 0.05$) had no significant positive effect on behavior intention to dine at green restaurants, whereas utilitarian value (β =0.136, t-value = 2.489, $p < 0.05$) had a significant positive effect on behavior intention to dine at green restaurants. Accordingly, the results support Hypothesis 4 but not Hypothesis 3. Further, the statistical results show that preference significantly positively impacts behavior intention of a green restaurant (β = 0.713, t-value = 20.293, $p < 0.001$) and explaining 62.8% of the variance in behavior intention. As a result, Hypothesis 5 is supported.

This study adopted PLS to test the hypotheses. The paths of the relationships among hedonic value, utilitarian value, consumer preferences and behavior intention are shown in Figure 2.

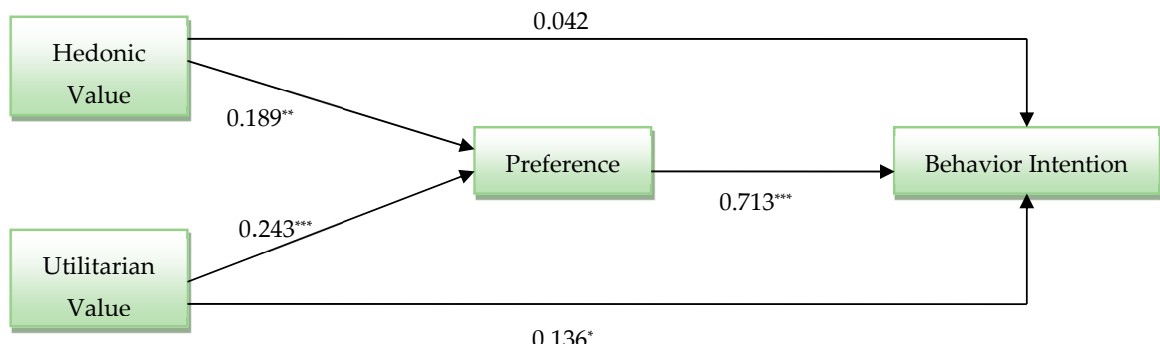

**Figure 2.** PLS results of the structural model. Note: * $p < 0.05$; ** $p < 0.01$; *** $p < 0.001$.

*4.4. Further Analysis*

We also discuss the mediating effect of preferences on the relationships between utilitarian value and behavior intentions. First, utilitarian value significantly positively affects preferences (β = 0.349, $p < 0.001$). Second, utilitarian value significantly positively influences behavior intentions (β = 0.427, $p < 0.001$). Third, preferences significantly positively affect behavior intentions (β = 0.771, $p < 0.001$). Finally, when preference is controlled, the influence of utilitarian value on behavioral intention will be significantly reduced, but not zero (Table 5). Therefore, preference can be regarded as a partial mediating variable for the effect of utilitarian value on behavior intentions, because the previously observed effect of utilitarian value on behavior intentions was significantly reduced.

**Table 5.** Mediators between preferences and behavior intentions.

| Independent Variable | Preference | Behavior Intention (BI) | |
|---|---|---|---|
| | | Step I | Step II |
| Utilitarian | 0.362 *** | 0.427 *** | 0.170 *** |
| Preference | — | — | 0.710 *** |
| $R^2$ | 0.131 | 0.182 | 0.620 |
| F-statistics | 41.565 *** | 61.451 *** | 224.145 *** |

Note: Numbers denote the beta coefficient of a particular variable. *** denotes significance at the 0.001 levels.

A Sobel test was carried out [61] in order to support the findings on mediating processes proposed by Baron and Kenny [62]. The Sobel test results showed that preference significantly mediates the relationship between utilitarian value and behavior intention (t-value = 3.375, $p < 0.001$).

## 5. Discussion

The purpose of this study was to examine the relationships among hedonic and utilitarian values, customer preference and behavioral intentions of the green restaurants. In summary, PLS analysis shows that the proposed model can well predict consumers' behavioral intentions to visit green restaurants and positively talk about their preferences for restaurants, indicating its applicability in the hospitality industries, especially the green restaurant business. The following findings are generated from this work.

First, according to our data analysis of supporting Hypotheses 1 and 2, it was observed that consumers do view utilitarian and hedonic values to be critical in shaping their preferences for green restaurants although utilitarian value was found to be a stronger predictor than hedonic value while customers' preferences play a crucial role in changing behavior intentions, echoing prior studies [19,41].

Second, our results also indicate that utilitarian value plays a greater role in behavior intentions as shown by prior studies [22,24,26,30]. That means the experience of dining at a green restaurant may be more appropriately characterized as a strong goal-oriented and instrumental event, rather than as an essentially pleasurable activity. For example, green restaurant managers should proudly advertise and honor their unique food offerings. This will in turn strengthen consumers' perceptions of the utilitarian value of green restaurants, further promoting preferences for and patronage of green restaurants.

Third, we also recommend that restaurants should strive to enhance the practical value of green restaurants and avoid putting customers off for major functional or beneficial reasons. These views echo those of Yu et al. [11]. When customers choose to dine at a green restaurant, they care about the quality of their food. In regard to green foods, customers seek appealing ingredients, a variety of sustainable menu items and fresh materials.

Fourth, the empirical results of this article also reveal that hedonic value does not significantly influence behavior intentions to dine at green restaurants, diverging from the results of prior studies [22,24,26,30]. This finding may be rooted in the fact that while consumers may prefer a green restaurant that offers a delightful and pleasant environment, these offerings are not sufficient to encourage consumers to dine at a green restaurant. As another possible interpretation, the impact of the hedonic value on behavior intentions to dine at green restaurants may be fully mediated by preferences.

## 6. Implications

This paper introduces a practical framework to support academics and practitioners. Academically, the study makes the contribution to the body of knowledge. This study investigated the influence of perceived values (hedonic and utilitarian values) on consumer preference and behavior intentions of the green restaurant. Previous studies have mainly investigated on levels of equality among green restaurants [11,18] and consumers' perceptions and beliefs about green consumption [4,14–16]. The present study is the first to investigate the hedonic and utilitarian values

effect on consumer preference and behavior intentions toward the green restaurant. Furthermore, the results also discovered that the effects of utilitarian value on behavioral intentions are mediated by preferences to dine at green restaurants. Thus, this study provides a pioneer reference for similar studies in the future.

For a practical implication point of view, this research evidences that the utilitarian value plays a greater role in consumers' preference and behavior intentions. As the significant role, marketing for green restaurants should focus on promoting utilitarian value. Enhancing green restaurants' utilitarian value in the eyes of consumers should be the central mission in marketing strategies. In turn, consumer may patronize green restaurants to satisfy their functional needs for good health, which are at the heart of the utilitarian value. Green restaurateurs should clearly convey to consumers their use of sustainable, healthy, organic or local food sources. We expect our study to help green restaurant managers better understand consumers' rationales for dining at green restaurants and to respond accordingly, thereby eventually allocating their resources to provide hedonic and utilitarian value to customers. This should shape customers' preferences and behavioral intentions, which should in turn affect patronage behavior.

## 7. Limitations and Future Research

This study is not without limitations. First, we collected data for Taiwan. Cultural differences in patronage intentions observed for Taiwanese green restaurants and green restaurants in other countries have not been examined. In the future, researchers may draw such comparisons. Second, the data employed for this study were collected via convenience sampling. Despite the frequent use of this form of data collection in previous studies, the approach limits the degree of generalization possible [20,24,63,64]. In the future, researchers can attempt to use other sampling methods (for example, random sampling, systematic sampling, etc.) to illustrate the regular distribution of the population's characteristics. Third, this study focuses on green restaurants, so other types of restaurants are not investigated. Hence, other types of restaurants may be studied in future studies for further generalization and comparisons.

**Author Contributions:** K-S.W. and Y-M.T. formulated the study, analyzed the collected data and wrote the article. All authors read and approved the final manuscript.

**Funding:** This research was supported by the Ministry of Science and Technology of Taiwan, R.O.C. under Grant no. MOST-103-2410-H-032-064.

**Conflicts of Interest:** The authors declare that there is no conflict of interest.

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
