# Peer review of "Sustainability Development in Hospitality: The Effect of Perceived Value on Customers’ Green Restaurant Behavioral Intention"

_sustainability, doi:10.3390/su11071987_

Reviewer 1 Report

This is an interesting paper that could be potentially publishable subject to some revisions that are discussed in more detail below.

Detailed comments

Research design - Methods:

Provide more details on the sample selection.

It is essential to explain how you have eliminated the possibility of sample selection bias.

Results: Discuss on the generalization of the results of the study.

Implications: Discuss on both methodological and practical implications of the study.

Author Response

Dear reviewer,

Please find our response attached.

Reviewer 2 Report

This is an interesting article on a topic of significant interest and it is my pleasure to review it.

The manuscript is meticulous and impressive in its approach but, at the same time, interesting and innovative, managing to fill a gap in literature on these topics (sustainability, green economy, food industry), of great importance in the economy and society.

The paper has merit; however, I have some considerations and suggestions for improving the quality of the article.

- The abstract is sketchy, schematically written, especially in its first part (related to the context and motivation of the study). We encourage authors to write more elaborate phrases and pay more attention to the style of ideas’s layout. In many cases, the interest of readers is shaped at the first contact with the paper, i.e. by the title and abstract.

- The paper advances several hypotheses. Although the analysis is laborious and carefully conducted, in the final part (interpretation and discussion), there is a lack of a summary of the results, related to the proposed assumptions. We recommend that the authors explicitly address (in a few short paragraphs) how the hypotheses are accepted / rejected / unresponsive or they need to be redrafted / re-evaluated. This would serve the purpose of the paper and, while allowing conclusions to be drawn and launching the future researches’ directions.

-  Hypotheses 3 and 4 are stated somewhat unclearly ... consumers’ behavior intentions to engage in green restaurant patronage …  In our opinion, it does not reveal much of the essence of the question and how much these hypotheses (H3 and H4) differ from the meaning of the previous hypotheses (H1 and H2). Perhaps expressions like engage ... patronage may suggest other meanings than (the presumed ones) of the present study. Brief additional explanations (of course, without changing the meaning and drafting of hypotheses) could contribute to a clearer presentation of these issues.

Thank you for the opportunity to review this article and good luck!

Author Response

(The authors gave the same response as above.)

Reviewer 3 Report

Introduction is detailed and interesting. I recommend to add brief description of used methods as well as each section (what will be in the article) to the introduction. 

Additionally, it is worth to extend the discussion in the paper. The Authors introduce only one - two sentences where they confront their results with findings by other researchers. In my opinion discussion should be added as separate section or as "results and discussion".  

I do not have more comments. The article is an advanced scientific study. IT has a well thought out structure, a good theoretical basis and interesting empirical research developed using advanced research methods. Congratulations!

Author Response

(The authors gave the same response as above.)
